# 9.2%-efficient core-shell structured antimony selenide nanorod array solar cells

Zhiqiang Li[1], Xiaoyang Liang[1], Gang Li[1], Haixu Liu[1], Huiyu Zhang[1], Jianxin Guo[1], Jingwei Chen[1], Kai Shen[2], Xingyuan San[1], Wei Yu[1], Ruud E.I. Schropp [2] & Yaohua Mai[2]

Antimony selenide ($Sb_2Se_3$) has a one-dimensional (1D) crystal structure comprising of covalently bonded $(Sb_4Se_6)_n$ ribbons stacking together through van der Waals force. This special structure results in anisotropic optical and electrical properties. Currently, the photovoltaic device performance is dominated by the grain orientation in the $Sb_2Se_3$ thin film absorbers. Effective approaches to enhance the carrier collection and overall power-conversion efficiency are urgently required. Here, we report the construction of $Sb_2Se_3$ solar cells with high-quality $Sb_2Se_3$ nanorod arrays absorber along the [001] direction, which is beneficial for sun-light absorption and charge carrier extraction. An efficiency of 9.2%, which is the highest value reported so far for this type of solar cells, is achieved by junction interface engineering. Our cell design provides an approach to further improve the efficiency of $Sb_2Se_3$-based solar cells.

[1] Hebei Key Laboratory of Optic-Electronic Information Materials, College of Physics Science and Technology, Hebei University, Baoding 071002, China.
[2] Institute of New Energy Technology, College of Information Science and Technology, Jinan University, Guangzhou 510632, China. These authors contributed equally: Zhiqiang Li, Xiaoyang Liang, Gang Li. Correspondence and requests for materials should be addressed to Z.L. (email: lizhiqiang@hbu.edu.cn) or to Y.M. (email: yaohuamai@jnu.edu.cn)

Among inorganic semiconductor thin film photovoltaics, cadmium telluride (CdTe) and copper indium gallium selenide (Cu(In,Ga)Se$_2$) solar cells have reached power-conversion efficiencies of over 22%[1,2]. The high device performance is possible due to the enough photon absorption, high bulk lifetime, superior carrier collection efficiency, and excellent junction interface. The chalcogenide antimony selenide (Sb$_2$Se$_3$) recently emerged as a promising alternative light-absorber material for high-efficiency photovoltaic devices due to its attractive properties, such as a single phase structure, proper optical bandgap (1.1–1.3 eV), high light absorption coefficient ($10^5$ cm$^{-1}$ at around 600 nm), low toxicity, and high element abundance[3–8]. The use of the chalcogenide Sb$_2$Se$_3$ avoids the issue of low In and Ga availability. The application of Sb$_2$Se$_3$ in photovoltaic devices as light-absorber was explored by Nair et al. in 2000s, yielding a rather low conversion efficiency of 0.66%[9,10]. Since the notable efficiency values of 3.21% and 2.26% obtained in 2014 by Choi et al. and Zhou et al., respectively, Sb$_2$Se$_3$-based solar cells have experienced rapid development[3,11]. A power-conversion efficiencies of 6.0% was reported for a zinc oxide (ZnO)/Sb$_2$Se$_3$ heterojunction and 6.5% for a cadmium sulfide (CdS)/Sb$_2$Se$_3$ heterojunction with PbS quantum dot film as hole-transporting layer, respectively[12,13]. Moreover, a 7.6% efficiency was reported this year, due to an improved crystallinity of Sb$_2$Se$_3$ thin film absorbers[14]. However, for Sb$_2$Se$_3$ to become a low cost, high abundancy compound to replace Cu(In,Ga)Se$_2$, this value is still too much behind that of state-of-the-art Cu(In,Ga)Se$_2$ solar cells. We here present a concept based on growing Sb$_2$Se$_3$ nanorod arrays that can lead to fundamentally improved solar cells. This method thus far had led to cells with a certified efficiency of 9.2%.

One attractive feature of Sb$_2$Se$_3$ is that it has a one-dimensional (1D) crystal structure and highly anisotropic properties. The Sb$_2$Se$_3$ crystal consists of ribbon-like (Sb$_4$Se$_6$)$_n$ units linked through van der Waals forces in the [010] and [100] direction, while strong covalent Sb–Se bonds make the units holding together in the [001] direction[3,15]. This apparently direction-dependent bonding nature will result in significant anisotropy. Theoretical calculation revealed that the surfaces parallel to the [001] direction, such as (110), (120) surfaces, have lower formation energies than the other surfaces and were terminated with surfaces free of dangling bonds[15]. Moreover, theoretical calculations and experimental results exhibited that carrier transport in the [001] direction is much easier than that in other directions[15,16]. Thus, the devices are expected to offer appealing photoresponse and device performance if the Sb$_2$Se$_3$ absorber consists of (Sb$_4$Se$_6$)$_n$ ribbons stacked vertically on the substrate. However, up to date, only quality [221]-oriented absorbers have been fabricated, in which the (Sb$_4$Se$_6$)$_n$ ribbons were tilted and have a certain degree with the substrate. On the other hand, the optimal Sb$_2$Se$_3$ absorber thickness for these devices were limited to the range of 0.3–0.6 μm due to the electron diffusion length ($L_e$) of only 0.3 μm in the [221] direction[16]. Due to this effect, the higher electron diffusion length $L_e$ along the [001] direction, which approaches 1.7 μm (five times that along the [221] direction[16]), could thus far not be fully exploited.

In this work, we address this limitation and grew Sb$_2$Se$_3$ nanorod arrays and solar cells with [001]-orientation on Mo-coated glass substrates using the close spaced sublimation (CSS) technique. A growth model is presented to investigate the mechanism covering the stages from atom absorption at the Mo surface to growth of the thin film structure towards the formation of aligned 1D Sb$_2$Se$_3$ nanorod arrays. We investigated the junction structure of the CdS/Sb$_2$Se$_3$ nanorod interface. We here reveal the migration of element antimony (Sb) into the whole CdS buffer layer if no specific precautions are taken. Subsequently, we

introduce a very thin titanium oxide (TiO$_2$) layer deposited by atomic layer deposition (ALD) technique at the CdS/Sb$_2$Se$_3$ junction interface. The interface engineering with TiO$_2$ leads to an independently verified record power-conversion efficiency of 9.2% for the Sb$_2$Se$_3$ solar cells (ZnO:Al/ZnO/CdS/TiO$_2$/Sb$_2$Se$_3$ nanorod arrays/MoSe$_2$/Mo) with an absorber thickness over 1000 nm while maintaining a high fill factor of 70.3%. The values of external quantum efficiency (EQE) are higher than 85% in a wide spectral range from 550 to 900 nm, approximating the values of well-developed CdS/Cu(In,Ga)Se$_2$ thin film solar cells. This work can facilitate the preparation and application of patterned 1D Sb$_2$Se$_3$-based nanostructures for applications in sensor arrays, piezoelectric antenna arrays, and other electronic and optoelectronic devices.

## Results

**Characterization of Sb$_2$Se$_3$ nanorod arrays.** It is worth noting that, to our knowledge, the fabrication of high quality ribboned Sb$_2$Se$_3$ nanorod arrays on Mo-coated glass substrate by the CSS technique has not been previously reported. The surface and cross-sectional morphologies of the as-deposited Sb$_2$Se$_3$ nanorod arrays were characterized by scanning electron microscope (SEM) in Fig. 1a, b, respectively. A high density array of Sb$_2$Se$_3$ nanorods grown vertically on the substrate with diameters ranging from 100 to 300 nm and lengths of about 1200 nm was observed. The crystal structure and phase purity of the Sb$_2$Se$_3$ nanorod arrays were measured by X-ray diffraction (XRD) as depicted in Fig. 1c. The arrays exhibit the orthorhombic crystal geometry belonging to the space group of *Pbnm* (JCPDS 15-0861) with no detectable impurities of other phases. It is important to note that only strong (*hk1*) and (*hk2*) diffraction peaks are observed in the XRD pattern, suggesting that the Sb$_2$Se$_3$ nanorod arrays have a preferred orientation along the *c*-axis direction. The intensity ratios of $I_{101}/I_{221}$ and $I_{002}/I_{221}$ for the nanorod arrays reached 0.42 and 0.73, respectively. These ratios are much higher than those of thin films with the (221)-preferred orientation in previous reports[12,17]. Since the (221)-oriented grain consists of (Sb$_4$Se$_6$)$_n$ ribbons grown vertically to the substrate with a tilt angle, the increased $I_{101}/I_{221}$ and $I_{002}/I_{221}$ values hint that the Sb$_2$Se$_3$ nanorod arrays are grown with enhanced preference along the *c*-axis [001] direction and at a higher tilt angle between (Sb$_4$Se$_6$)$_n$ ribbons and the substrate, compared to the (221)-oriented thin films[4,15]. We further relied on high-resolution transmission electron microscopy (HRTEM) to reveal the crystal orientation of the individual Sb$_2$Se$_3$ nanorods. Samples were cross-sectioned by focused ion beam and a TEM image of the nanorod array is shown in Fig. 1d. The interplanar d-spacings of 0.389 nm and 0.521 nm correspond to the (001) and (210) planes of orthorhombic Sb$_2$Se$_3$, respectively, as shown in Fig. 1e, which is consistent with the 1D single-crystalline Sb$_2$Se$_3$ nanostructures synthesized by chemical synthesis methods[18,19]. The corresponding selected-area electron diffraction (SAED) pattern (Fig. 1f) exhibited the vertical relationship of the (001) and (210) planes, indicating the [1$\bar{2}$0] crystallographic axis of the *Pbnm* space group and the Sb$_2$Se$_3$ nanorod, suggesting that the Sb$_2$Se$_3$ nanorod arrays in this work grow along the [001] direction. Analysis on additional Sb$_2$Se$_3$ nanorods further supported that the Sb$_2$Se$_3$ nanorod arrays were grown along the [001] direction (Supplementary Figure 1). The SAED characterization provides a direct observation of the atomic arrangement of the Sb$_2$Se$_3$ nanorod and echoes previous XRD and SEM results.

**Growth model of Sb$_2$Se$_3$ nanorod arrays on Mo substrate.** As shown in Fig. 2, a series of plan-view and cross-sectional SEM images of Sb$_2$Se$_3$ grown with different durations on Mo substrate

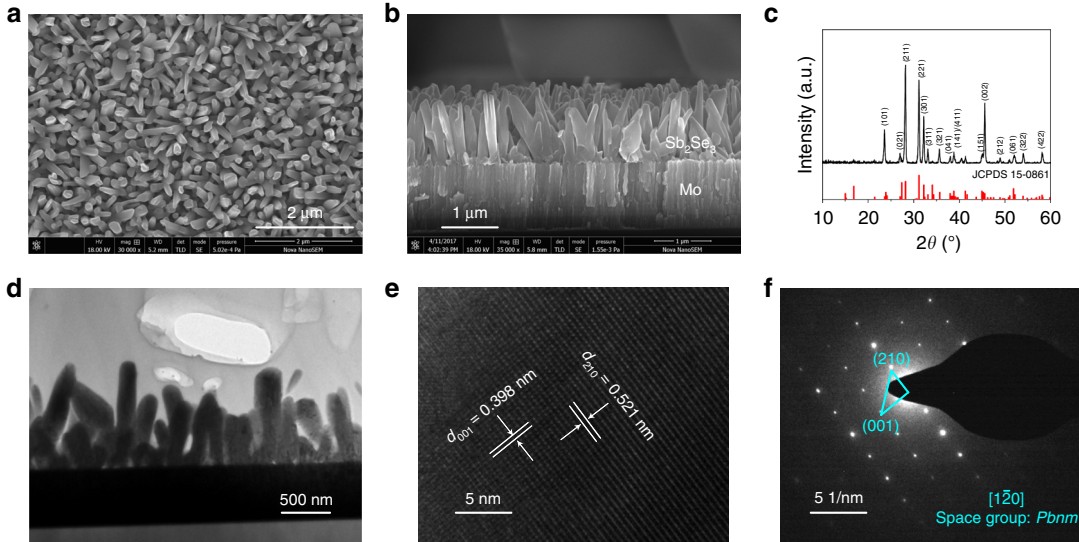

**Fig. 1** Microscopy and spectroscopy of Sb$_2$Se$_3$ nanorod array. **a–c** Top-view (**a**), cross-sectional (**b**), SEM images and X-ray diffraction pattern (**c**) of the Sb$_2$Se$_3$ nanorod arrays grown on Mo-coated glass substrate. **d–f** TEM image (**d**), high resolution TEM (HRTEM) image (**e**), and the corresponding selected-area electron diffraction (SAED) pattern (**f**) of the Sb$_2$Se$_3$ nanorod array

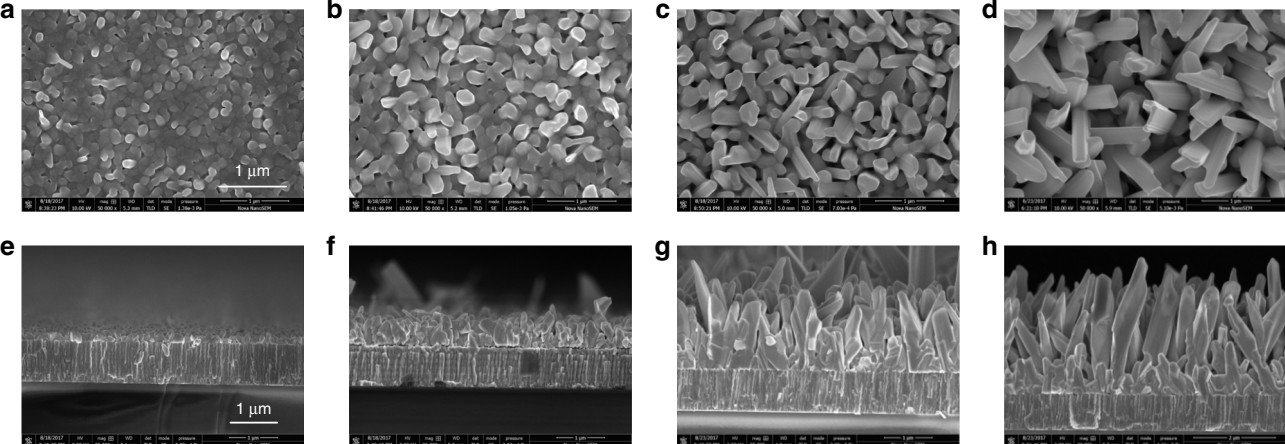

**Fig. 2** Morphology evolution of Sb$_2$Se$_3$: from thin film to nanorod array. **a–d** Top-view SEM images of Sb$_2$Se$_3$ with different deposition times, **a** 60 s, **b** 120 s, **c** 160 s, and **d** 180 s. **e–h** The corresponding cross-sectional images of Sb$_2$Se$_3$ with different deposition time, **e** 60 s, **f** 120 s, **g** 160 s, and **h** 180 s. The scale bar for **a–h** is 1 μm

exhibit the morphological evolution of Sb$_2$Se$_3$. It was found that with increasing growth durations from 60 to 180 s the morphologies of Sb$_2$Se$_3$ samples vary from a compact thin film structure to an aligned nanorod array structure. As seen from the corresponding cross-sectional images, the thickness of the Sb$_2$Se$_3$ layer was 200, 600, 1000 and 2000 nm for the samples grown for 60, 120, 160 and 180 s, respectively. It indicates that both thickness and growth rate are increased as the deposition proceeds. The CSS-processed Sb$_2$Se$_3$ is a smooth and compact film composed of grains with uniform grain size of about 100 nm in the first 60 s (Fig. 2a, e). When the growth time increases to 120 s, the grain size increases to 200–300 nm and the sample still displays film structure morphology, though the surface becomes porous and some craters can be observed (Fig. 2b, f). For the sample grown for 160 s (Fig. 2c, g), it is observed that the Sb$_2$Se$_3$ consists of a compact bottom layer and a nanorod-array top layer vertical to the substrate. The vertical nanorod array appears to grow on top of the compact bottom layer. As the growth times increases further, to 180 s, the thickness of the top nanorod-array layer

increases while the compact bottom layer thickness shrinks (Fig. 2d, h).

Based on the above observation, we propose a model to understand the mechanism governing the transition in the growth process from Sb$_2$Se$_3$ thin film to nanorod array. The growth process of Sb$_2$Se$_3$ can be divided into four stages: surface absorption, film growth, splitting, and nanorod array growth stage. For the first (surface absorption) stage, we have generated an atomistic model shown in Fig. 3a based on the following considerations: first, Sb$_2$Se$_3$ possesses a 1D crystal structure and is comprised of (Sb$_4$Se$_6$)$_n$ ribbons. Considering the combination between the (Sb$_4$Se$_6$)$_n$ ribbon and the substrate surface, we calculated the atom displacement distributions by the Vienna ab initio Simulation Package (VASP)[20]. The calculated results show that the Sb and Se atoms are dispersed from Sb$_4$Se$_6$ and scattered on the Mo surface and that the ribboned structure of Sb$_4$Se$_6$ collapses if the Sb$_4$Se$_6$ unit runs parallel to the Mo (110) surface (Supplementary Figure 2 and Supplementary Figure 3a). On the contrary, when the Sb$_4$Se$_6$ unit is standing vertically on the Mo

(110) plane, the simulated results display that the unit is stable with lower distortion (Supplementary Figure 3b). Second, despite the decomposition of $Sb_2Se_3$ during the thermal process, the absorption of Sb or Se atoms at the $Sb_4Se_6$/Mo interface is also taken into account. The degree of lattice deformation for the $Sb_4Se_6$/Mo, $Sb_4Se_6$/Sb/Mo and $Sb_4Se_6$/Se/Mo absorption models, respectively, is 0.755, 0.642 and 0.534. This indicates that on the Mo surface the absorption of one Se atom layer prior to $(Sb_4Se_6)_n$ ribbons is favored rather than the vertical growth of $(Sb_4Se_6)_n$ ribbons (Supplementary Figure 3c, 3d and Supplementary Table 1).

During film growth, splitting, and nanorod growth stages, the $Sb_2Se_3$ grains grow bigger as $Sb_2Se_3$ vapor continuously evaporates from the $Sb_2Se_3$ source, and then the transition from thin film to nanorod growth occurs when the generated lateral stress beyond the tolerance of the van der Waals forces between the $(Sb_4Se_6)_n$ ribbons in the deposited $Sb_2Se_3$ films. The nanorods get longer and more in number and the splitting goes deeper into the film as the growth time proceeds (Fig. 3d), which could be attributed to the higher growth rate in the ribbon direction due to the stronger covalent Sb–Se bonds internally in the ribbon.

**Device performance and characterization.** To investigate the effect of different absorber morphologies on the performance of the $Sb_2Se_3$ solar cells, the devices were finished by successively depositing the CdS buffer, high-resistance (HR) and low-resistance (LR) ZnO layer, and front Ag contact. The devices were divided into three groups according to the thicknesses and morphologies of the CSS-processed $Sb_2Se_3$ absorbers. For description clarity, we denoted the $Sb_2Se_3$ thin film absorbers with thickness between 200 and 600 nm as TF-$Sb_2Se_3$, the $Sb_2Se_3$ thickness in the range of 650 to 1100 nm, comprising a double layer (vertical nanorod-array top layer and compact-film bottom layer) as M-$Sb_2Se_3$, and $Sb_2Se_3$ absorbers thicker than 1100 nm with nearly an entire nanorod-array structure as NA-$Sb_2Se_3$.

Figure 4a displays representative current density-voltage (J-V) curves of the solar cells employing the TF-$Sb_2Se_3$, M-$Sb_2Se_3$, and NA-$Sb_2Se_3$ absorbers, respectively. Typical J-V characterizations performed under standard test conditions (STC) yielded an optimal conversion efficiency of 4.78% for the M-$Sb_2Se_3$ solar cell with an open circuit voltage ($V_{OC}$) of 0.370 V, short circuit current density ($J_{SC}$) of 27.43 mA cm$^{-2}$, and fill factor (FF) of 47.46% (see Table 1). The NA-$Sb_2Se_3$ samples show substantially

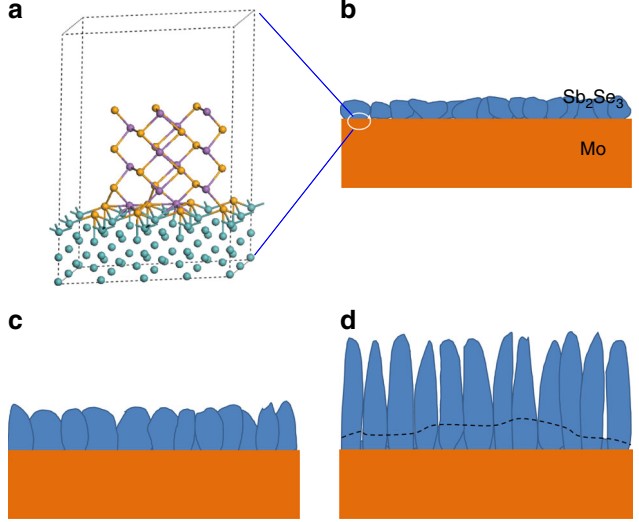

**Fig. 3** Growth model of the $Sb_2Se_3$ nanorod arrays on Mo substrate. **a** Atomistic model of $Sb_4Se_6$ unit on the (110) plane of Mo. **b–d** Schematics of the $Sb_2Se_3$ at different growth stages, **b** thin film growth, **c** split, and **d** nanorod array growth (top part exhibits obvious nanorod array morphology and bottom is compact layer)

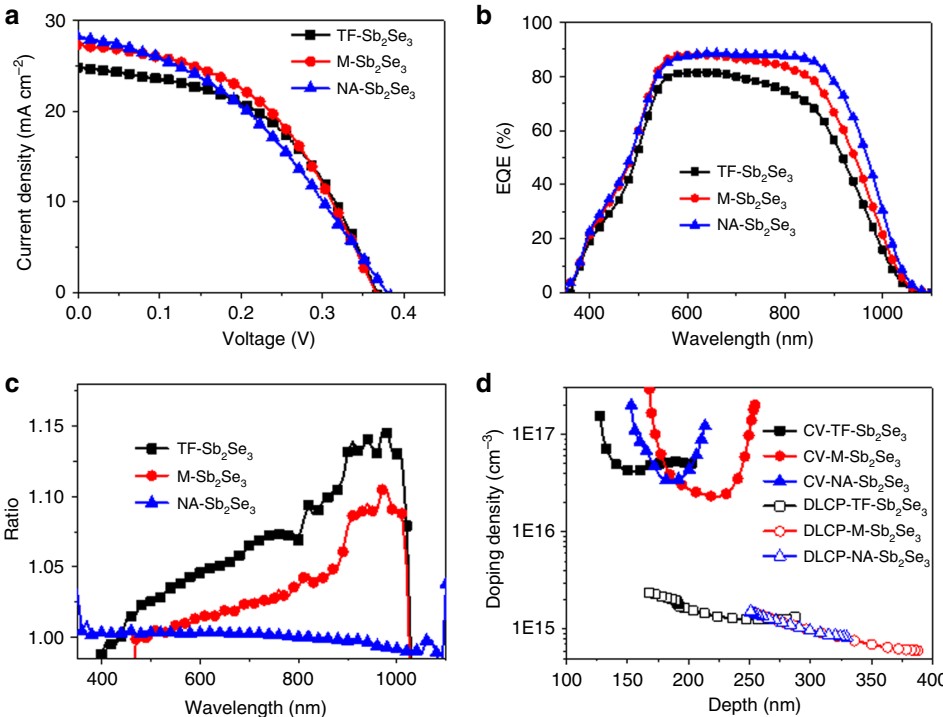

**Fig. 4** Device performances of solar cells with different absorbers. **a** The representative J-V curves of the solar cells with different $Sb_2Se_3$ absorbers. **b**, **c** EQE spectra and the ratio of EQE(−0.5 V)/EQE(0 V) curves of the solar cells. **d** C-V profiling and DLCP profiling of the solar cells

**Table 1 Photovoltaic performance parameters for the $Sb_2Se_3$ solar cells with different absorber structure (Fig. 4a)**

| Absorber | $V_{OC}$ (V) | $J_{SC}$ (mA cm$^{-2}$) | Fill factor | Efficiency (%) |
|---|---|---|---|---|
| TF-$Sb_2Se_3$ | 0.368 | 24.87 | 49.53 | 4.53 |
| M-$Sb_2Se_3$ | 0.370 | 27.34 | 47.46 | 4.78 |
| NA-$Sb_2Se_3$ | 0.382 | 28.60 | 40.18 | 4.39 |

reduced FF, which may be attributed to strong CdS/$Sb_2Se_3$ interface recombination.

The presence of the nanorod structure in the absorbers increases the $J_{SC}$ of the solar cells, which is mainly due to the enhanced long wavelength response (Fig. 4b). The rough surface of the thicker nanorod absorber enhances the light harvesting and thus reduces the optical reflection (Supplementary Figure 4)[21]. At the same time, the [001] preferential orientation of the nanorods facilitates long-range carrier transport along the $(Sb_4Se_6)_n$ ribbons and thus guarantees carrier extraction and high $J_{SC}$[16]. This is also supported by the EQE and biased EQE results. As shown in Fig. 4b, the EQE spectrum of the TF-$Sb_2Se_3$ device reaches a maximum value of 80% at about 550 nm, then declines both at shorter and longer wavelength due to the strong absorption of the CdS buffer and the insufficient generation and/or collection of carriers at the back side, respectively. This observation is consistent with previous reports of $Sb_2Se_3$-based thin film solar cells[22,23]. For the M-$Sb_2Se_3$ device, the maximum value of EQE reaches 88% at approximately 550 nm, higher than that of the TF-$Sb_2Se_3$ device, partly due to its lower reflectance. The EQE spectra of the NA-$Sb_2Se_3$ device demonstrates a relatively wide EQE plateau with values approaching 87% between 550 and 900 nm, and a gradual decrease towards longer wavelengths. EQE spectra were also measured under bias-voltage conditions ($-0.5$ V), and the curves describing the ratio of EQE ($-0.5$ V) over EQE (0 V) are shown in Fig. 4c. For the NA-$Sb_2Se_3$ device, the EQE ratio is approximately unity over the whole spectral range, while that of the TF-$Sb_2Se_3$ and M-$Sb_2Se_3$ devices is strongly bias dependent, especially at long wavelength. This indicates that the photo-generated carriers in the latter devices are not collected completely and the collection requires an internal electric field. The high and wide plateau and its weak bias-voltage dependence of the EQE spectrum of NA-$Sb_2Se_3$ device reveals that the carrier collection is highly efficient for the $Sb_2Se_3$ nanorod array structure along the [001] direction, explaining the higher $J_{SC}$ value of the NA-$Sb_2Se_3$ device compared to that of the M-$Sb_2Se_3$ device.

We then turned to the issue of the junction properties of the TF-$Sb_2Se_3$, M-$Sb_2Se_3$ and NA-$Sb_2Se_3$ solar cells. In order to understand their AC behavior, an equivalent circuit model was introduced. It consists of serial conductance, junction conductance and the capacitance element, which mainly includes the junction interface and trapping state induced capacitance. (Supplementary Figure 5). The junction capacitance is frequency independent while trapping capacitance is strongly frequency-dependent[24–26]. In comparison with the TF-$Sb_2Se_3$, the M-$Sb_2Se_3$ and NA-$Sb_2Se_3$ device exhibit smaller and less frequency dependent capacitances, indicating that the growth of $Sb_2Se_3$ nanorods reduces the defect density in the $Sb_2Se_3$ absorber or at its surface.

We further performed the capacitance–voltage (C-V) profiling and deep-level capacitance profiling (DLCP) measurements on these devices for characterizing the defects. In general, the C-V measurement is relation to free carriers, junction interface defects and bulk defects, while DLCP measurement is less sensitive to the junction interface defects[27]. As shown in Fig. 4d, the $N_{DLCP}$

values for these three devices are in the range of $4 \times 10^{14}$ to $2 \times 10^{15}$ cm$^{-3}$, which are lower than the values obtained for reference samples of $Sb_2Se_3$ grown on ZnO or $TiO_2$ layer as well as for $Sb_2Se_3$ deposited by thermal evaporation on Mo substrate ($4.6 \times 10^{15}$ to $1.1 \times 10^{17}$ cm$^{-3}$)[12,17,28]. This suggests that the CSS-processed $Sb_2Se_3$ absorbers on Mo substrate have a lower bulk defect density. On the other hand, $N_{DLCP}$ for TF-$Sb_2Se_3$ device was a little higher than that for M-$Sb_2Se_3$ and NA-$Sb_2Se_3$ devices, indicating the reduced bulk defect densities due to the evolution of $Sb_2Se_3$ from thin films to nanorod array structure. However, the $N_{CV}$ values were much higher than the $N_{DLCP}$ values for these three devices, indicating serious interface defects present at the CdS/$Sb_2Se_3$ interface. The depletion width ($W_d$) is mainly located in the $Sb_2Se_3$ region at the CdS/$Sb_2Se_3$ junction interface since the doping density of CdS is much higher than that of the $Sb_2Se_3$ absorbers[16,29,30]. Hence, the interfacial defect density could be calculated to be $2.77 \times 10^{12}$ cm$^{-2}$, $2.85 \times 10^{12}$ cm$^{-2}$ and $3.21 \times 10^{12}$ cm$^{-2}$ for TF-$Sb_2Se_3$, M-$Sb_2Se_3$ and NA-$Sb_2Se_3$ devices, respectively. These values are higher than those of CdS/$Sb_2Se_3$, ZnO/$Sb_2Se_3$ or $TiO_2$/$Sb_2Se_3$ in superstrate configurations, indicating that much more interface state activity can be expected for CBD-CdS buffer grown on $Sb_2Se_3$ absorbers[12].

**CdS/$Sb_2Se_3$ junction interface**. To explore the coverage of CBD-CdS layer coated on the $Sb_2Se_3$ nanorod surface and the inter-diffusion of elements at the CdS/$Sb_2Se_3$ interface, we employed SEM, TEM, and high-angle annular dark-field scanning transmission electron microscope (HAADF-STEM) equipped with energy-dispersive spectroscopy (EDX) to characterize the interface of our CdS-coated $Sb_2Se_3$ nanorod array samples. As shown in Fig. 5a, b, the CBD growth procedure yields a uniform, dense, and pin-hole free CdS film, and the CdS layer completely covers the $Sb_2Se_3$ nanorod array surface, yielding a CdS/$Sb_2Se_3$ core-shell structure. The morphology of the CdS film reveals a fine-grain accumulated structure. The TEM image (Fig. 5c, d) displays that the thickness of the CdS coated at the top of the $Sb_2Se_3$ nanorods is about 50 to 60 nm. More details on the CdS film growth on the $Sb_2Se_3$ nanorod arrays reveal that the CBD-CdS is not only present on top of the nanorods but also penetrates into the space between the nanorods and conformally coats the sidewalls of the nanorods and in the valleys on the bottom compact layer, making the cell at least partially a radial junction cell. The uniform and complete coverage of CdS layer suggests good adhesion and well defined junction formation between the $Sb_2Se_3$ nanorod array and the CdS buffer layer.

A rectangular area in the Z-contrast HAADF cross-sectional image at the CdS/$Sb_2Se_3$ nanorod interface was chosen to analyze the Sb, Se, Cd, and S element distribution. As shown in Fig. 5e, element spatial mapping of Se, Cd, and S shows sharp edges, indicating negligible interfacial inter-diffusion of these three elements. On the contrary, the Sb element mapping exhibits an obvious two-zone behavior in the CdS/$Sb_2Se_3$ nanorod interface region, suggesting Sb-diffusion into the CBD-CdS layer. This phenomenon is quite different from the superstrate CdS/$Sb_2Se_3$ heterojunction case, in which the Cd, S, Sb, and Se elements mix together to form a thin n-type inter-diffusion layer and a buried homojunction at the interface, dictating charge separation and device performance in superstrate CdS/$Sb_2Se_3$ thin film solar cells. The presence of Sb in the whole CBD-processed CdS buffer layer can be attributed to the dissolution of $Sb_2Se_3$ in the alkaline precursor solution (Supplementary Table 2). During the CBD process, some ammonia was added into the precursor solution to supply a suitable environment for the chemical reactions, and thus it also reacted with the precursor to form surface growth complexes[31,32]. For reference, the metal chalcogenide was

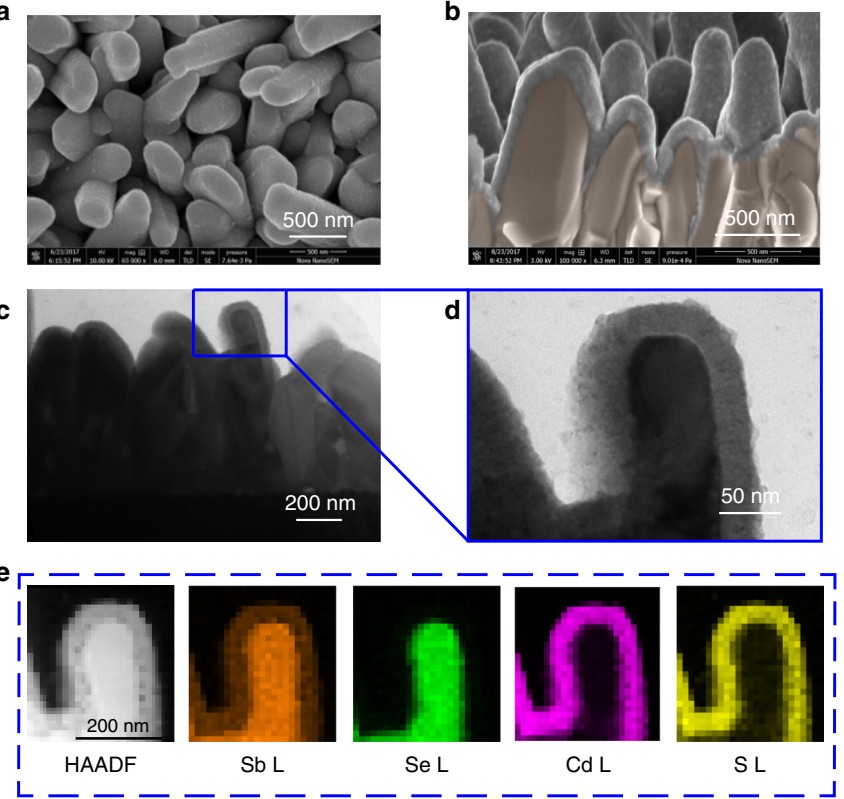

**Fig. 5** Characterization of CdS/Sb$_2$Se$_3$ junction interface. **a, b** Top-view (**a**) and cross-sectional (**b**) SEM images of CdS buffer deposited on Sb$_2$Se$_3$ nanorod arrays. **c–e** TEM (**c**, **d**) and HAADF-STEM image and energy-dispersive spectroscopy elemental mapping (**e**) of the CdS/Sb$_2$Se$_3$ junction interface. Elements detected: Sb L, Se L, Cd L, and S L

dissolved in hydrazine or ammonia sulfide solution through the formation of highly soluble metal chalcogenide complexes at a molecular level[33–35]. A similar dissolution process is expected to occur in the reaction of Sb$_2$Se$_3$ with NH$_4^+$ during the deposition of the CdS layer in an ammonia solution.

**Surface modification of Sb$_2$Se$_3$ nanorod arrays by thin ALD-TiO$_2$.** In order to address the issue of Sb diffusion and the concomitant high interface defect density, a very thin atomic layer deposited (ALD) TiO$_2$ layer was introduced between the Sb$_2$Se$_3$ nanorod array absorber and the CdS buffer to protect the Sb$_2$Se$_3$ from directly contacting the NH$_4^+$ ions during the deposition of the CdS layer by CBD method. The EDX line scan analysis shows that the Sb content in the CdS layer was reduced for the CdS shell grown on ALD-TiO$_2$ modified Sb$_2$Se$_3$ nanorod (Supplementary Figure 6). The decrease of Sb content in the CdS shell indicated that the thin ALD-TiO$_2$ could efficiently reduce the dissolution of Sb$_2$Se$_3$ during the CBD process. Moreover, the corrosion rate of Sb$_2$Se$_3$ layer in the ammonia solutions is slightly decreased after performing 20 cycles of TiO$_2$ (Supplementary Figure 7 and Supplementary Table 2). Figure 6b, c exhibit the top-view and cross-sectional images of the Sb$_2$Se$_3$ solar cells after applying all steps to a successfully completed fabrication. The device exhibits a stamp-like nanopatterned surface morphology and fewer holes and gaps are observed in the cross-sectional image, suggesting that the CBD-processed CdS and sputtered ZnO/ZnO:Al completely covers the top of the Sb$_2$Se$_3$ nanorods as well as the lower parts within the space between nanorods.

Figure 6d displays the J-V curve of our best device in this work under simulated AM1.5 G solar illumination. This device was fabricated with 20 cycles of ALD TiO$_2$ on the Sb$_2$Se$_3$ absorber

prior to the deposition of the CdS buffer. The cell exhibits a $V_{OC}$ of 0.40 V, a $J_{SC}$ of 32.58 mA cm$^{-2}$, a FF of 70.3%, resulting in an overall power-conversion efficiency of 9.2%, which has independently been verified by National Institute of Metrology of China (Supplementary Figure 8). A histogram of the device efficiencies obtained from 100 individually fabricated devices is shown in Fig. 6f. The average $V_{OC}$, $J_{SC}$, FF, and conversion efficiency were 399 ± 33 mV, 29.80 ± 3.36 mA cm$^{-2}$, 64.46 ± 12.01% and 7.69 ± 1.56%, respectively. Figure 6e depicts the corresponding EQE spectrum for the champion solar cell. It exhibits a broad plateau of over 85% between 550 nm and 900 nm and the integrated current density reaches a value as high as 31.48 mA cm$^{-2}$. The photoresponse in the plateau region is higher than that of Sb$_2$Se$_3$ solar cells in a superstrate configuration and is comparable with that of CdS/CIGS thin film solar cells prepared in our laboratory with an efficiency of 15%, as shown in Supplementary Figure 9. Nonetheless, there is large current loss at wavelengths below 550 nm due to strong parasitic absorption of the CdS buffer since the electron-hole pairs generated in the CdS layer are not collected. Therefore, it is desirable to replace the CdS with another wide band gap buffer material for further optimization.

Compared with the device without ALD-TiO$_2$ (Fig. 4, Supplementary Figure 10), the enhancement in conversion efficiency mainly results from an increase in $V_{OC}$ and FF, which is tentatively attributed to the reduction of dissolution of Sb$_2$Se$_3$ during the CBD process and/or the reduction of shunt paths by the ALD-TiO$_2$ of the surface defects on the Sb$_2$Se$_3$ nanorods (the dangling bonds at the tips of the (Sb$_4$Se$_6$)$_n$ nanoribbons). As shown in Supplementary Figure 11, Kelvin probe force microscope (KPFM) was employed to study the surface properties of the Sb$_2$Se$_3$ nanorod array surfaces before and after the deposition

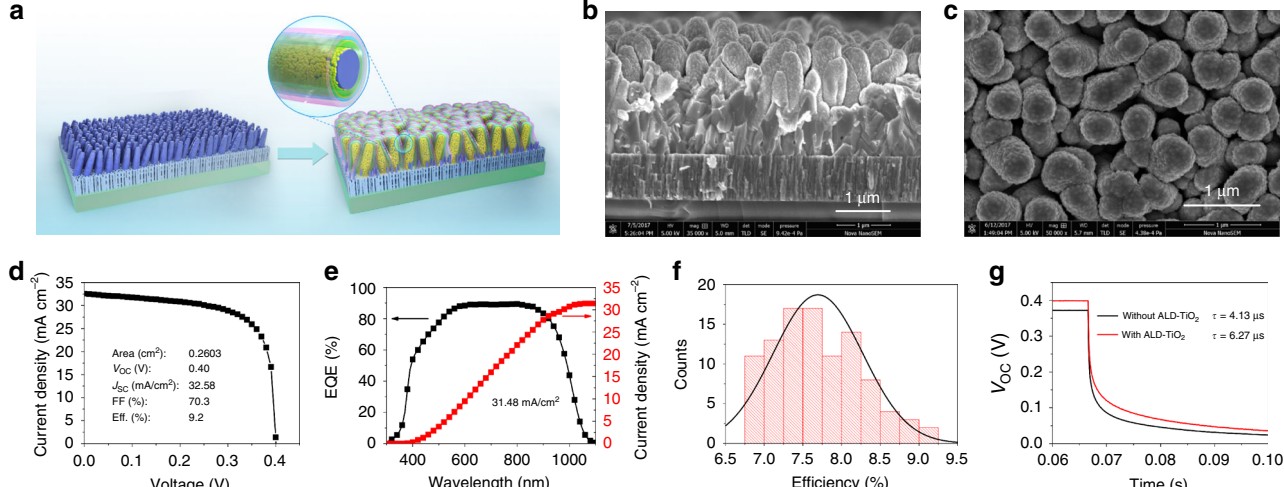

**Fig. 6** Solar cell structure and mechanistic investigation of ALD-TiO$_2$ on Sb$_2$Se$_3$ nanorod arrays. **a** Schematic of the Sb$_2$Se$_3$ nanorod arrays on Mo-coated glass and finished Sb$_2$Se$_3$/CdS core/shell nanorod array solar cells. **b, c** Cross-sectional (**b**) and top view (**c**) SEM images of the completed CdS/Sb$_2$Se$_3$ solar cells. **d, e** J-V curve (**d**) and EQE spectrum (**e**) of the champion device (area = 0.2603 cm$^2$). **f** Histogram of device efficiency over 100 individually fabricated solar cells. **g** V$_{OC}$ decay curves of the solar cells with and without ALD-TiO$_2$ layer

of 20 cycles of TiO$_2$. While the average roughness stays at the same value (100 nm), the average surface potential difference decreases from 28.8 to 10.4 mV after the deposition of the thin ALD-TiO$_2$. This suggests that a thin layer of ALD-TiO$_2$ improves the surface band bending at the side walls and reduces the surface defects at the tips of the Sb$_2$Se$_3$ nanorods[15,36]. A surface potential difference of 280 mV was observed between the Sb$_2$Se$_3$ layer before and after thin ALD-TiO$_2$ modification. Taking into account of the valence band maximum (VBM) and band gap of Sb$_2$Se$_3$, we obtained the energy level diagram of the CdS/(TiO$_2$) Sb$_2$Se$_3$ interface (Supplementary Figure 12). The conduction band minimum (CBM) of Sb$_2$Se$_3$ layer is shifted by about 0.13 eV towards to the vacuum level after ALD-TiO$_2$ modification. The downshifted of the CBM could decrease the conduction band offset at buffer/absorber interface, and lead to the increased fill factor. Furthermore, the possible shunt paths for the CdS/Sb$_2$Se3 junction with and without ALD-TiO$_2$ were detected by conductive atomic force microscopy (C-AFM). For the sample without ALD-TiO$_2$ some white dots, representing the detected current, are observed (Supplementary Figure 13), indicating the poor coverage of CdS and the presence of shunt leakage due to local discontinuity or pinholes in the CdS buffers. On the contrary, with the insertion of thin ALD-TiO$_2$ between the CdS buffer and the Sb$_2$Se$_3$ nanorod array absorber, the white dotted area decreases or even vanishes, suggesting reduced shunt leakage.

The ALD-TiO$_2$ layer may also passivate the surface defects of the Sb$_2$Se$_3$ layer. This can be confirmed by the V$_{OC}$ decay measurement, which is related to the carrier recombination rate and the carrier lifetimes. Figure 6g displays the V$_{OC}$ decay curves of two representative Sb$_2$Se$_3$ solar cells, with and without ALD-TiO$_2$ thin layer. The cell with 20 cycles of ALD-TiO$_2$ layer exhibits an obvious longer decay time than the cell without ALD-TiO$_2$. Furthermore, as the thin ALD-TiO$_2$ layer is compact and has excellent film conformity due to its layer-by-layer growth, it is expected to reduce or even prevent the chemical reaction of Sb$_2$Se$_3$ with the growth solution during the CBD deposition of the CdS buffer layer, leading to a more pure CdS buffer layer. The influence of doping of Sb in CdS buffer layers has not been exclusively demonstrated thus far and requires more investigation in the near future. We investigate the stability of the Sb$_2$Se$_3$

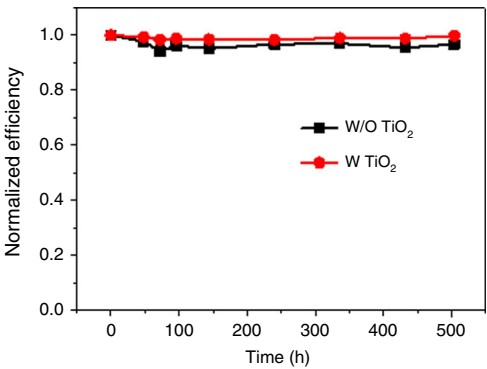

**Fig. 7** Device stability. Stability of representative Sb$_2$Se$_3$ solar cells without and with TiO$_2$ modification

nanorod array based solar cells. As shown in Fig. 7, the normalized efficiency of the CdS/Sb$_2$Se$_3$ solar cell with TiO$_2$ modification hold a slightly higher value (~97% of its initial value) than that of the device without TiO$_2$ modification (~94% of its initial value) after storage in air for more than 500 h.

## Discussion

In summary, we have demonstrated the fabrication of high quality solar cells employing a 1D Sb$_2$Se$_3$ nanorod array absorber with a height of more than 1000 nm in the substrate configuration. TEM analysis indicated that the growth of nanorods is along the [001] direction. We propose a split growth model based on the morphology evolution from the thin film to a nanorod array. The solar cells exhibited excellent EQE spectra in the whole working wavelength range (higher than 85% between 550 and 900 nm), indicating that there is long-range carrier transport along the [001] direction. Furthermore, we found that Sb diffuses into the CdS buffer due to the solubility of Sb$_2$Se$_3$ in the alkaline solution during the CBD process. A very thin TiO$_2$ layer deposited by ALD was introduced prior to the deposition of CdS buffer layer, leading to an improved V$_{OC}$, FF as well as conversion efficiency. This cell design and these results provide important

progress towards the understanding and application of 1D-structured $Sb_2Se_3$ crystals.

## Methods

**Solar cell fabrication.** The bilayer Mo back contacts were prepared by a two-step magnetron sputtering process, which consisted of high working pressure (2.0 Pa) and low working pressure (0.3 Pa) process. The total thickness of Mo was about 1000 nm. A Mo selenization process was carried out at 620°C for 20 min to form about 20 nm thick $MoSe_2$ layer prior to the deposition of $Sb_2Se_3$[17]. The $Sb_2Se_3$ absorber layers were grown on selenized Mo-coated glass by using a homemade CSS system. In CSS system, the thermocouple was inserted into the graphite plate to directly detect the temperatures of substrate and evaporation source, respectively. The temperatures of $Sb_2Se_3$ source and substrate holder were controlled by two sets of lamp heaters and thermocouples. The distance between the source and the sample holder was 11 mm. We started the deposition when the pressure was below $10^{-2}$ Pa. First, the source and sample holder were warmed up to 480 °C and 270 °C, respectively, in 200 seconds, and maintained at the high temperatures for hundreds of seconds to obtain the desired $Sb_2Se_3$ absorber thickness. The thicknesses of the $Sb_2Se_3$ layers in the range of 200–2000 nm were controlled by adjusting the duration ranging from 60–180 s at high temperature. The samples were taken out after cooling down to about 150 °C in about 1 h. After that, the $Sb_2Se_3$ samples were coated with 60 nm of CdS by chemical bath deposition at a bath temperature of 70 °C. Window layers of HR and LR ZnO films were sputtered from pure ZnO and ZnO:Al targets ($Al_2O_3$ 2 wt%-doped). Top Ag grids of the solar cells were finally formed by thermal evaporation. The complete $Sb_2Se_3$ solar cells have a structure of glass/Mo/$MoSe_2$/$Sb_2Se_3$/HR-ZnO/LR-ZnO/Ag. $TiO_2$ was deposited at 150 °C in a homemade ALD reactor system, which using titanium isopropoxide (TTIP) and $H_2O$ as Ti and O precursors, respectively. One deposition cycle involves a $H_2O$ pulse of 0.5 s, a $N_2$ pulse of 60 s, a TTIP pulse of 0.5 s, and 60 s of $N_2$ purging, and each deposition cycle was started with a $H_2O$ pulse and terminated with a TTIP pulse. About 2 nm thickness of $TiO_2$ coating was deposited in 20 cycles.

**Material and device characterization.** SEM observations were performed on a FEI Nova NANOSEM 450 field-emission microscope and the TEM measurements were carried out on a FEI Tecnai G2 transmission electron microscope. The optical properties were recorded using a Perkin-Elmer Lambda 950 spectrophotometer. The XRD data were collected with a Bruker D8 Advance diffractometer. The current density-voltage (*J-V*) measurement was performed using an AM1.5 solar simulator equipped with a 300 W Xenon lamp (Model No. XES-100S1, SAN-EI, Japan). The EQE was measured by an Enlitech QER3011 system equipped with a 150 W xenon light source. Capacitance-voltage (*C-V*) measurement was performed on Agilent B1500A Semiconductor device analyzer in the dark at room temperature. Carrier-lifetime measurements were performed using the DN-AE01 Dyenamo toolbox with a white light-emitting diode (Luxeon Star 1W) as the light source[37,38].

**Simulation methods.** All calculations of $Sb_2Se_3$ growing on the Mo (110) were calculated by the VASP. The DFT calculations employed the Perdew-Burke-Ernzerhof (PBE) generalized gradient approximation (GGA) exchange-correlation functional and the projector-augmented wave (PAW) method. An energy cut-off of 500 eV was applied for the plane wave expansion of the wave functions. $2 \times 4 \times 1$ Monkhorst-pack mesh for k-point sampling are required to relaxation all models of the $Sb_2Se_3$ sheet growing on the Mo (110) with or without Se and Sb layers.

## Data availability

The data supporting this study are available from the authors on request.

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

## Acknowledgements

This work was supported by the Advanced Talents Incubation Program of the Hebei University (801260201001), National Natural Science Foundation of China (NSFC No.61804040), Scientific Research Foundation for the Returned Overseas Chinese Scholars (CG2015003004), and Natural Science Foundation of Hebei Province (No. E2016201028).

## Author contributions

Z.L. and Y.M. conceived the idea and designed the experiments. Z.L., X.L. and G.L. performed most of the device fabrication and characterization. H.L., H.Z. and W.Y. conducted the $TiO_2$ deposition. K.S. and X.S. assisted in the TEM and EDX mapping characterization and data analysis. J.G. and J.C. carried out the theoretical simulation and analyzed the results. Z.L., R.E.I.S., and Y.M. analyzed the overall results and wrote the paper. Y.M. supervised the project and all authors discussed the experiments and commented on the manuscript.

## Additional information

**Competing interests:** The authors declare no competing interests.

