## [Peer Review File · Nature Communications]

Reviewers' comments:

Reviewer #1 (Remarks to the Author):

In this submission, Zhiqiang Li et. al. reported the fabrication of [001] oriented Sb₂Se₃ nanorod array, and then applying a ALD derived TiO₂ passivation layer, and finally achieved a certified 9.2% efficiency with substrate device configuration. 9.2% is a new record for Sb₂Se₃ solar cells, a new thin film photovoltaics with simple and low-toxic composition, low-cost fabrication and outstanding stability. The author also carefully characterize their Sb₂Se₃ thin films and devices to present a convincing explanation. This paper is well written and reported a great breakthrough in Sb₂Se₃ thin film photovoltaics, and also provided an avenue for further improvement with efficiency >10%. This is a very important paper and I strongly support its publication in Nature Communications. A few minor points that need to be addressed:

1. Please provide more details about the film deposition and device fabrication, particularly for those minor details that dramatically affect device performance, either positively or negatively. For example, how accurate is temperature control of the home-made system? How long does it take to cool down? How the device is stored before the TiO₂ and CdS deposition? How TiO₂ is made? Ti-terminated or O-terminated? The goal of publication is to make others could reproduce the results; more details are welcome by the readers.
2. Please provide the storage and operation stability data for this type of device. Generally a TiO₂ sandwich layer should improve the operational stability of Sb₂Se₃ solar cells with CdS buffer layer. If possible, please provide long term stability (>500 hours).
3. The author tried to explain the improved FF and provided some useful information. How about the shunt and series resistance? Please provide these values for devices with and without TiO₂ layer (for high FF and low FF devices).
4. Is the efficiency total area efficiency or active area efficiency? What is the variation of the device parameters observed? What is highest Voc, Jsc and FF even observed? How is the reproducibility?
5. Etching of Sb₂Se₃ film by ammonium from the CBD solution is not surprising. The authors are encouraged to provide more results to fully understand this process. For example, why S is removed and Sb is incorporated into CdS? The authors proposed the dissolution-re-deposition mechanism; if this is true, Sb species should be detected in the CBD solution.

Reviewer #2 (Remarks to the Author):

This work reports an impressive improvement in efficiency (up to 9.2%) for Sb₂Se₃ based solar cells. This result has brought the efficiency of the Sb₂Se₃ solar cell to a comparable level of CZTSe. It appears they might face the same issue of low open circuit voltage, which is possibly due to the defects in the absorber layer. However, it is still in a relatively early stage. Perhaps further optimization could improve this shortcoming.

The authors claim the improvement in the Sb₂Se₃ orientation is important. The real effect is at best secondary, because Sb₂Se₃ nanorods are not really straight, e.g., with strong (101) scattering, and this improvement alone does not yield much improvement in efficiency. The major improvement perhaps comes from the inserting of an ALD-TiO₂ layer between the Sb₂Se₃ absorber and CdS layer. I understand it has two effects: reducing the interdiffusion of Sb into CdS, thus, reducing defects; and preventing shunt. The general idea is not necessarily new, thus, the novelty claim is questionable or unnecessary. It has been used in other systems, such as Si/ZnO cells.

Fig. 2(e) shows the elemental distribution without the TiO₂ layer. The authors should provide comparison with TiO₂ to support the conclusion.

The TiO₂ layer will have effect on the charge transport. The authors should provide the band-diagrams of the devices with and without the TiO₂ layer to explain the underlying physics.
I support the publication of this work after above mentioned revisions.

Reviewer #3 (Remarks to the Author):

The authors have investigated unidimensional antimony selenide (Sb₂Se₃) as an active material in solar cells. The structure 1D reduced traps density improving the charge carrier collection and an increment in power efficiency was observed. The model was proposed to grow nanorods arrays on Mo substrates.

The work presented is disorder, using characterization techniques without a congruent sequence. First, SEM to show nanorods arrays after solar cells to know optoelectronics properties in solar cell, capacitance measurements and DLCP profiling to support the increment in power efficiency, characterization of the interface CdS/Sb₂Se₃, solar cells measurements with TiO₂ deposited by ALD, finally Two-dimensional topography and AFM images of the CdS/Sb₂Se₃ junctions. The manuscript is not clear and easy to follow.

Important claims need to be supported as improve collection efficiency, some measurements are required, interface recombination in CdS/Sb₂Se₃ junction, what kind of carrier recombination is predominant in this interface? extended explanation is required about how the capacitance change with frequency and traps density. Based on the above. I do not recommend this manuscript to be accepted in Nature Communications.

Reviewers' comments:

Reviewer #1 (Remarks to the Author):

In this submission, Zhiqiang Li et. al. reported the fabrication of [001] oriented Sb₂Se₃ nanorod array, and then applying a ALD derived TiO₂ passivation layer, and finally achieved a certified 9.2% efficiency with substrate device configuration. 9.2% is a new record for Sb₂Se₃ solar cells, a new thin film photovoltaics with simple and low-toxic composition, low-cost fabrication and outstanding stability. The author also carefully characterize their Sb₂Se₃ thin films and devices to present a convincing explanation. This paper is well written and reported a great breakthrough in Sb₂Se₃ thin film photovoltaics, and also provided an avenue for further improvement with efficiency >10%. This is a very important paper and I strongly support its publication in Nature Communications. A few minor points that need to be addressed:

1. Please provide more details about the film deposition and device fabrication, particularly for those minor details that dramatically affect device performance, either positively or negatively. For example, how accurate is temperature control of the home-made system? How long does it take to cool down? How the device is stored before the TiO₂ and CdS deposition? How TiO₂ is made? Ti-terminated or O-terminated? The goal of publication is to make others could reproduce the results; more details are welcome by the readers.

Reply: Thank you so much for your comment. We have added the requested details about the film deposition and device fabrication in experimental section of the revised manuscript.

Revision: More experimental details are added on page 14 of the revised manuscript: The additions are: "The thermocouples were inserted into the graphite plate of the substrate holder and the evaporation source holder to directly detect the temperatures of substrate and evaporation source, respectively.", "The samples were taken out after cooling down to about 150°C in about 1 hour.", and "each deposition cycle was started with a H₂O pulse and terminated with a TTIP pulse."

2. Please provide the storage and operation stability data for this type of device. Generally a TiO₂ sandwich layer should improve the operational stability of Sb₂Se₃ solar cells with CdS buffer layer. If possible, please provide long term stability (>500 hours).

Reply: We measured device stability for both of the un-encapsulated Sb₂Se₃ solar cells with and without thin ALD-TiO₂ surface modification. As shown in Figure 7, the solar cell without TiO₂ revealed minor performance degradation, with its conversion efficiency after 500 hours of storage time dropped to about 94% of its initial value. In contrast, the device with ALD-TiO₂ modification exhibited higher stability and remained above 97% during the entire 500 hours of testing.

Revision: In the revised manuscript, Figure 7 is added and a discussion is added on page 13 of the main text.

Figure 7:

Figure 7 Device stability. Efficiency versus storage time of representative Sb_2Se_3 solar cells without and with TiO_2 modification.

Added discussion: We investigated the stability of the Sb_2Se_3 nanorod array based solar cells. As shown in Figure 7, the normalized efficiency of a typical un-encapsulated $\text{CdS}/\text{Sb}_2\text{Se}_3$ solar cell with TiO_2 modification retains a slightly higher value (~97% of its initial value) than that of the device without TiO_2 modification (~94% of its initial value) after storage in air (humidity 30-40%) for more than 500 hours.

3. The author tried to explain the improved FF and provided some useful information. How about the shunt and series resistance? Please provide these values for devices with and without TiO_2 layer (for high FF and low FF devices).

Reply: The comparison of the shunt and series resistance for the Sb_2Se_3 solar cells with and without TiO_2 are now shown in Supplementary Figure 12. The solar cells modified by thin ALD- TiO_2 exhibit smaller series resistance and larger shunt resistance compared with the device without ALD- TiO_2 .

Revision: Supplementary Figure 12 is added in the revised Supplementary Information.

4. Is the efficiency total area efficiency or active area efficiency? What is the variation of the device parameters observed? What is highest V_{oc} , J_{sc} and FF even observed? How is the reproducibility?

Reply: The efficiency is the total area efficiency. The variation of the device performance parameters (V_{oc} , J_{sc} , FF and efficiency) is shown in Figure R1 below. The highest values for V_{oc} , J_{sc} and FF are 429 mV, 32.58 mA/cm^2 and 76.67%, respectively. The device efficiencies of about 8 batches are shown in Figure R2 below, and the batches show good reproducibility.

Figure R1 The variation of the device performance parameters for the Sb_2Se_3 solar cells with TiO_2 modification.

Figure R2 Device efficiency distribution for 8 batches.

5. Etching of Sb_2Se_3 film by ammonium from the CBD solution is not surprising. The authors are encouraged to provide more results to fully understand this process. For example, why S is removed and Sb is incorporated into CdS? The authors proposed the dissolution-re-deposition mechanism; if this is true, Sb species should be detected in the CBD solution.

Reply: Yes, the element Sb was indeed detected in the final CBD solution. After the deposition of CdS buffer layers, we collected the final solution and checked the element Sb and Se by using inductively coupled plasma-atomic emission spectrometry (ICP-AES Agilent ICPOES 730). As shown in Supplementary Table 2, the content of Sb and Se was 0.0699 mg/L and 0.0793 mg/L, respectively. Furthermore, the Sb_2Se_3 absorber layer with thickness of about 1000 nm was found to be totally dissolved after immersion into the CBD solution for more than 2 hours, and the Sb and Se content increased to 0.7562 mg/L and 0.5463 mg/L, respectively.

Revision: In the revised manuscript, Supplementary Table 2 of ICP test results from CBD solution to check the presence of element Sb and Se in the solution was added to the Supplementary

Information .

Reviewer #2 (Remarks to the Author):

This work reports an impressive improvement in efficiency (up to 9.2%) for Sb₂Se₃ based solar cells. This result has brought the efficiency of the Sb₂Se₃ solar cell to a comparable level of CZTSe. It appears they might face the same issue of low open circuit voltage, which is possibly due to the defects in the absorber layer. However, it is still in a relatively early stage. Perhaps further optimization could improve this shortcoming.

The authors claim the improvement in the Sb₂Se₃ orientation is important. The real effect is at best secondary, because Sb₂Se₃ nanorods are not really straight, e.g., with strong (101) scattering, and this improvement alone does not yield much improvement in efficiency. The major improvement perhaps comes from the inserting of an ALD-TiO₂ layer between the Sb₂Se₃ absorber and CdS layer. I understand it has two effects: reducing the interdiffusion of Sb into CdS, thus, reducing defects; and preventing shunt. The general idea is not necessarily new, thus, the novelty claim is questionable or unnecessary. It has been used in other systems, such as Si/ZnO cells.

Fig. 2(e) shows the elemental distribution without the TiO₂ layer. The authors should provide comparison with TiO₂ to support the conclusion.

Reply: Thank you very much for your suggestion. We performed EDX line scans to analyze the Sb distribution of CdS/Sb₂Se₃ nanorod without and with ALD-TiO₂ modification (Supplementary Figure 6). In the CdS/Sb₂Se₃ nanorod, two shoulders in the Sb distribution, located at 30% and 40% relative content, were observed in the CdS shell. In comparison, only one smaller shoulder in the Sb distribution, located at 20% relative content, was detected in the CdS shell for the CdS/TiO₂ modified Sb₂Se₃ nanorod. The decrease of Sb content in the CdS shell indicates that the thin ALD-TiO₂ can efficiently reduce the dissolution of Sb₂Se₃ during the CBD process.

Revision: In the revised manuscript, a discussion was added on page 11 of the main text, as follows: "The EDX line scan analysis shows that the Sb content in the CdS layer was reduced for the CdS shell grown on ALD-TiO₂ modified Sb₂Se₃ nanorod (Supplementary Figure 6). The decrease of Sb content in the CdS shell indicates that the thin ALD-TiO₂ can efficiently reduce the dissolution of Sb₂Se₃ during the CBD process. Moreover, the corrosion rate of Sb₂Se₃ layer in the ammonia solutions is slightly decreased after performing 20 cycles of TiO₂ (Supplementary Figure 7 and Supplementary Table 2).", and Supplementary Figure 6 and more discussions are added on page 7 of the Supplementary Information.

The TiO₂ layer will have effect on the charge transport. The authors should provide the band-diagrams of the devices with and without the TiO₂ layer to explain the underlying physics. I support the publication of this work after above mentioned revisions.

Reply: Thank you so much for your comment. We detected the band energy of Sb₂Se₃ absorbers

via KPFM and XPS, in order to ascertain if there is any change of the energy levels of the Sb_2Se_3 absorbers before and after ALD- TiO_2 deposition. A surface potential increase of about 280 mV was obtained for the Sb_2Se_3 absorber layer after ALD- TiO_2 modification. Moreover, taking into account the VBM and band gaps of Sb_2Se_3 and CdS layers, we obtained the energy level diagram of the $\text{CdS}/(\text{TiO}_2)\text{Sb}_2\text{Se}_3$ interface (Supplementary Figure 11). The conduction band minimum (CBM) of Sb_2Se_3 layer was shifted by about 0.13 eV towards the Fermi level after ALD- TiO_2 modification. The downshift of the CBM decreases the conduction band offset at the buffer/absorber interface, and leads to the increased fill factor.

Revision: In the revised manuscript, a discussion is added on page 12 of the main text, as follows: "A surface potential difference of 280 mV was observed between the Sb_2Se_3 layer before and after thin ALD- TiO_2 modification. Taking into account the valence band maximum (VBM) and the band gaps of the Sb_2Se_3 and CdS layers, we obtained the energy level diagram of the $\text{CdS}/(\text{TiO}_2)\text{Sb}_2\text{Se}_3$ interface (Supplementary Figure 11). The conduction band minimum (CBM) of Sb_2Se_3 layer is shifted by about 0.13 eV after ALD- TiO_2 modification. The downshift of the CBM decreases the conduction band offset at the buffer/absorber interface, and leads to the increased fill factor.", and Supplementary Figure 11 is added.

Reviewer #3 (Remarks to the Author):

The authors have investigated unidimensional antimony selenide (Sb_2Se_3) as an active material in solar cells. The structure 1D reduced traps density improving the charge carrier collection and an increment in power efficiency was observed. The model was proposed to grow nanorods arrays on Mo substrates.

The work presented is disorder, using characterization techniques without a congruent sequence. First, SEM to show nanorods arrays after solar cells to know optoelectronics properties in solar cell, capacitance measurements and DLCP profiling to support the increment in power efficiency, characterization of the interface $\text{CdS}/\text{Sb}_2\text{Se}_3$, solar cells measurements with TiO_2 deposited by ALD, finally Two-dimensional topography and AFM images of the $\text{CdS}/\text{Sb}_2\text{Se}_3$ junctions. The manuscript is not clear and easy to follow.

Reply: Thank you so much for your comment. We have reorganized our works in the revised manuscript.

Revision: In the revised manuscript, the C-AFM images are replaced by the device stability test, some of the capacitance measurements are transferred to the Supplementary information. Moreover, more focused characterizations and discussions about junction interface properties are added for better clarity. We believe that the manuscript now gives a more logical and comprehensive account of the main findings.

Important claims need to be supported as improve collection efficiency, some measurements are required, interface recombination in $\text{CdS}/\text{Sb}_2\text{Se}_3$ junction, what kind of carrier recombination is predominant in this interface? extended explanation is required about how the capacitance

change with frequency and traps density. Based on the above. I do not recommend this manuscript to be accepted in Nature Communications.

Reply: In the section 'Device performance and characterization', we compared the EQE and biased EQE spectra for TF-Sb₂Se₃, M-Sb₂Se₃ and NA-Sb₂Se₃ solar cells in Figure 4b and 4c. The solar cell with NA-Sb₂Se₃ absorber has the best EQE values among these devices, which can be attributed to the reduced surface reflectance and enhanced collection length, which consists of the space-charge region width and the carrier diffusion length[1, 2]. Firstly, the NA-Sb₂Se₃ (Supplementary Figure 4, Sb₂Se₃-1100nm and Sb₂Se₃-2000 nm) have lower reflectance spectra than the TF-Sb₂Se₃ (Figure S4, Sb₂Se₃-200 nm) and M-Sb₂Se₃ (Supplementary Figure 4, Sb₂Se₃-600 nm) samples. Secondly, the NA-Sb₂Se₃ solar cell can be expected to have longer carrier diffusion length in [001] direction due to its [001]-preferred orientation. After all, previous works have demonstrated that the carrier diffusion length in [001] direction was about 5 times than that in [221] direction[3]. Moreover, the space-charge region width for solar cells is a function of the applied voltage bias. We demonstrated the ratio of biased EQE (EQE (-0.5 V)) over unbiased EQE (0 V) in Figure 4c. The EQE ratio for the NA-Sb₂Se₃ solar cells was less bias dependent, indicating that the space-charge width of the NA-Sb₂Se₃ solar cell did not limit the photogenerated carrier collection.

The biased EQE experiments thus support the claims regarding the collection efficiency.

Considering the influence of trap levels in the band gap, the equivalent circuit for the CdS/Sb₂Se₃ heterojunction solar cells is shown in Supplementary Figure 5a. The measured capacitance includes the contribution of junction and trap levels[1, 4]. Trap levels in the space charge region of a *pn* junction contribute to its admittance as follows[5, 6]: the traps are filled with electrons up to the Fermi level and interact with the nearest band edge by thermal capture and emission of carriers. During the measurement, a small dc voltage is usually applied to the junction to modulate the Fermi level with respect to the band edges and thereby modulate the occupancy of the trap states. Moreover, the filling and emptying of the traps take place at the emission rate, and the amount of trapped charges decrease when the trap occupancy can no longer follow the rapid jitter of the Fermi level; thus the trap capacitance decreases with increasing frequency. To separate the contribution of traps to the conductance *G*, the dc conductance of traps was subtracted from the measured values[7]. The frequency dependent normalized conductance $(G-G_d)/\omega$ was plot in Supplementary Figure 5b. No obvious peaks could be observed in the medium or high frequency region.

This hints that the defects might have a maximum value in the very low frequency region, which is beyond our testing facility capability. In addition, we could not obtain the defect distribution and determine whether the trap states were electron or hole traps from the frequency dependent junction capacitance measurements.

Revision: in the revised manuscript, Supplementary Figure 5 and one paragraph are added on page 5 and 6 of the Supplementary information.

[1] S.S. Hegedus, W.N. Shafarman, Thin-film solar cells: device measurements and analysis, Prog. Photovolt. Res. Appl., 12 (2004) 155-176.

- [2] W.N. Shafarman, R. Klenk, B.E. McCandless, Device and material characterization of Cu(InGa)Se₂ solar cells with increasing band gap, *J Appl. Phys.*, 79 (1996) 7324-7328.
- [3] C. Chen, D.C. Bobela, Y. Yang, S. Lu, K. Zeng, C. Ge, B. Yang, L. Gao, Y. Zhao, M.C. Beard, J. Tang, Characterization of basic physical properties of Sb₂Se₃ and its relevance for photovoltaics, *Front. Optoelectron.*, 10 (2017) 18-30.
- [4] P. Viktorovitch, G. Moddel, Interpretation of the conductance and capacitance frequency dependence of hydrogenated amorphous silicon Schottky barrier diodes, *J Appl. Phys.*, 51 (1980) 4847-4854.
- [5] T. Walter, R. Herberholz, C. Müller, H.W. Schock, Determination of defect distributions from admittance measurements and application to Cu(In,Ga)Se₂ based heterojunctions, *J Appl. Phys.*, 80 (1996) 4411-4420.
- [6] M. Luo, M. Leng, X. Liu, J. Chen, C. Chen, S. Qin, J. Tang, Thermal evaporation and characterization of superstrate CdS/Sb₂Se₃ solar cells, *Appl. Phys. Lett.*, 104 (2014) 173904.
- [7] J. Kneisel, K. Siemer, I. Luck, D. Braunig, Admittance spectroscopy of efficient CuInS₂ thin film solar cells, *J Appl. Phys.*, 88 (2000) 5474-5481.

REVIEWERS' COMMENTS:

Reviewer #1 (Remarks to the Author):

In the revised manuscript, the authors carefully and satisfactorily addressed all my previous concerns; I thus suggest direct acceptance of this manuscript without further revision.

Reviewer #2 (Remarks to the Author):

The authors have adequately addressed the comments of the reviewers. I recommend publication of this manuscript.

REVIEWERS' COMMENTS:

Reviewer #1 (Remarks to the Author):

In the revised manuscript, the authors carefully and satisfactorily addressed all my previous concerns; I thus suggest direct acceptance of this manuscript without further revision.

Reply: We thank the referee for his/her positive comments and previous advice. We thank the referee for the supporting comments on the relevance and importance of our work.

Reviewer #2 (Remarks to the Author):

The authors have adequately addressed the comments of the reviewers. I recommend publication of this manuscript.

Reply: We thank the referee for his/her positive comments.